# Nitrosyl-Heme and Heme Iron Intake from Processed Meats in Subjects from the EPIC-Spain Cohort

**DOI:** 10.3390/nu16060878

**Published:** 2024-03-18

**Authors:** Lucía Rizzolo-Brime, Andreu Farran-Codina, Ricard Bou, Leila Luján-Barroso, Jose Ramón Quirós, Pilar Amiano, María José Sánchez, Miguel Rodríguez-Barranco, Marcela Guevara, Conchi Moreno-Iribas, Alba Gasque, María-Dolores Chirlaque, Sandra M. Colorado-Yohar, José María Huerta Castaño, Antonio Agudo, Paula Jakszyn

**Affiliations:** 1Unit of Nutrition and Cancer, Catalan Institute of Oncology—ICO, 08908 L’Hospitalet de Llobregat, Barcelona, Spain; lrizzolo@idibell.cat (L.R.-B.); llujan@iconcologia.net (L.L.-B.); a.agudo@iconcologia.net (A.A.); 2Nutrition and Cancer Group, Epidemiology, Public Health, Cancer Prevention and Palliative Care Program, Bellvitge Biomedical Research Institute—IDIBELL, 08908 L’Hospitalet de Llobregat, Barcelona, Spain; 3Department of Nutrition, Food Science and Gastronomy, Faculty of Pharmacy, Institute of Nutrition and Food Safety (INSA-UB), University of Barcelona, Campus de l’Alimentació de Torribera, Av. Prat de la Riba 171, 08921 Santa Coloma de Gramenet, Barcelona, Spain; afarran@ub.edu; 4Food Safety and Functionality Program, Institute of Agrifood Research and Technology (IRTA), Finca Camps i Armet s/n, 17121 Monells, Girona, Spain; ricard.bou@irta.cat; 5Department of Public Health, Mental Health and Perinatal Nursing, Public Health, Mental Health and Perinatal Nursing, Universitat de Barcelona, Carrer de la Feixa Llarga s/n, 08907 L’Hospitalet de Llobregat, Barcelona, Spain; 6Public Health Directorate, 33001 Asturias, Spain; joseramon.quirosgarcia@asturias.org; 7CIBER Epidemiology and Public Health CIBERESP ISCIII, 28041 Madrid, Spain; p-amiano@euskadi.eus (P.A.); mdolores.chirlaque@carm.es (M.-D.C.); scyohar@gmail.com (S.M.C.-Y.); jmhuerta.carm@gmail.com (J.M.H.C.); 8Ministry of Health of the Basque Government, Sub Directorate for Public Health and Addictions of Gipuzkoa, 20013 San Sebastian, Guipúzcoa, Spain; 9Biodonostia Health Research Institute, Epidemiology of Chronic and Communicable Diseases Group, 20014 San Sebastian, Guipúzcoa, Spain; 10Escuela Andaluza de Salud Pública (EASP), 18011 Granada, Granada, Spain; mariajose.sanchez.easp@juntadeandalucia.es (M.J.S.); miguel.rodriguez.barranco.easp@juntadeandalucia.es (M.R.-B.); 11Instituto de Investigación Biosanitaria Ibs.GRANADA, 18012 Granada, Granada, Spain; 12Centro de Investigación Biomédica en Red de Epidemiología y Salud Pública (CIBERESP), 28029 Madrid, Spain; mp.guevara.eslava@navarra.es (M.G.); mc.moreno.iribas@navarra.es (C.M.-I.); 13Instituto de Salud Pública y Laboral de Navarra, 31003 Pamplona, Navarre, Spain; alba.gasque.satrustegui@navarra.es; 14Navarra Institute for Health Research (IdiSNA), 31008 Pamplona, Navarre, Spain; 15Department of Epidemiology, Murcia Regional Health Council-IMIB, 30008 Murcia, Murcia, Spain; 16Social-Health Department, Murcia University, 30008 Murcia, Murcia, Spain; 17Research Group on Demography and Health, National Faculty of Public Health, University of Antioquia, Medellín 050010, Colombia; 18Blanquerna School of Health Sciences, Ramon Llull University, 08022 Barcelona, Catalonia, Spain

**Keywords:** nitrosyl-heme, heme iron, nitrosylation, processed meat, dietary intake, meat derivatives

## Abstract

Background: The consumption of processed meats (PMs) and red meats are linked to the likelihood of developing colorectal cancer. Various theories have been proposed to explain this connection, focusing on nitrosyl-heme and heme iron intake. We hypothesized that differences in nitrosyl-heme and heme iron intakes will be associated with various sociodemographic and lifestyle factors. Methods: The study included 38,471 healthy volunteers (62% females) from five Spanish regions within the EPIC-Spain cohort. High-Performance Liquid Chromatography (HPLC) determined nitrosyl-heme and heme iron levels in the 39 most consumed PMs. Food intake was assessed using validated questionnaires in interviews. Nitrosyl-heme and heme iron intakes, adjusted for sex, age, body mass index (BMI), center, and energy intake, were expressed as geometric means due to their skewed distribution. Variance analysis identified foods explaining the variability of nitrosyl-heme and heme iron intakes. Results: The estimated intakes were 528.6 µg/day for nitrosyl-heme and 1676.2 µg/day for heme iron. Significant differences in nitrosyl-heme intake were found by sex, center, energy, and education level. Heme iron intake varied significantly by sex, center, energy, and smoking status. “*Jamón serrano*” and “*jamón cocido/jamón de York*” *had* the highest intake values, while “*morcilla asturiana*” and “*sangrecilla*” were key sources of nitrosyl-heme and heme iron. Conclusions: This is the first study to estimate levels of nitrosyl-heme intake directly in PMs for a large sample, revealing variations based on sex, BMI, smoking, and activity. Its data aids future exposure estimations in diverse populations.

## 1. Introduction

Intake of red meat and processed meat (PM) have been linked to different health outcomes, including increased risk of total mortality, type 2 diabetes, cardiovascular disease, and several cancer locations [1,2,3]. In 2015, the International Agency for Research on Cancer (IARC) [4] classified PMs as “carcinogenic” and red meat as “probably carcinogenic” for humans. In the same line, in 2018, the World Cancer Research Fund/American Institute for Cancer Research Continuous Update Project (WCRF/AIRC CUP) published their third expert report, which recommends that red meat consumption be limited to moderate amounts and processed meats to as little as possible, if at all in, following the recommendations in order to reduce cancer risk [3].

A series of different mechanisms have been proposed to explain the relationship between red/processed meat intake and cancer risk. Explanations have mainly been grounded on their content in mutagenic and carcinogenic compounds [5]. The fact that polycyclic aromatic hydrocarbons, heterocyclic amines and exogenously formed *N*-nitroso compounds have become increasingly significant in colorectal carcinogenesis [6] could support this view. In parallel, there is evidence arising from meta-analyses of epidemiological studies that suggest that heme iron consumption may explain the association between red meat intake and different types of cancer [7,8]. It is precisely the high content of heme, in the form of myoglobin, that confers meat its dark red color. In PMs, it is the nitrite present in the curing salt that causes nitrosylation of the heme molecule and then results in heme iron-nitrosyl being released from myoglobin during the curing and/or the cooking process [9]. Some studies have proposed that heme iron-nitrosyl may be potentially more toxic than the heme molecule itself [10,11], which could be one of the explanations for why PMs are more strongly associated with cancer when compared to fresh red meat. Additionally, several authors demonstrated that free iron, heme iron and nitrosyl-heme iron could induce the formation of reactive oxygen species and that this might finally trigger gene mutations and cellular damage [12,13,14]. 

Previous studies have analyzed nitrosylated and heme iron content but have been done for specific types of PMs only [15,16,17,18,19]. To the best of our knowledge, one single study has attempted to estimate the nitrosyl-heme intake in a population. They used data from the French Pork Institute to perform an indirect analysis, using the same fixed value to calculate the nitrosylated heme iron content of all PM items [20]. Compared to nitrosyl-heme, heme iron intake has been more extensively researched, though usually just expressed as a percentage of total iron available in meat items [21] or by using food composition tables [22]. However, available data on heme iron content in PMs are limited to some given PM items [23,24]. These reasons complicate the deep study of the role of carcinogenesis.

To better understand the impact of nitrosyl-heme and heme iron on human health and cancer risk, this study aimed to (a) estimate intake in subjects from the Spanish cohort of the European Prospective Investigation into Cancer and Nutrition (EPIC-Spain) based on laboratory determinations; (b) describe the differences according to sociodemographic and lifestyle variables and (c) identify those PM items that contributed the most to higher intake levels of nitrosyl-heme and heme iron in EPIC subjects.

## 2. Material and Methods

### 2.1. Study Population 

EPIC is a large prospective cohort study involving more than half a million participants (ages 25–70) recruited across 10 European countries between 1992 and 1999. This study was conducted within the framework of the EPIC-Spain cohort study [25]. The EPIC-Spain prospective cohort included 41,437 participants (62%, women) recruited between 1992 and 1996 from five Spanish regions: Asturias, the Gipuzkoa, Navarra, Murcia and Granada. In this study, 1097 subjects without dietary information or who had extreme or implausible caloric intake (the lowest and highest 1% of the ratio of total energy intake to energy requirement), as well as large consumers of PMs, were excluded. The final study sample consisted of 38,471 healthy volunteers (23,839 females and 14,632 males).

All participants gave their written informed consent, and the study was approved by the Ethical Review Board of each center.

### 2.2. Diet Information 

Dietary intake was assessed through a validated diet history questionnaire structured by meals administered face-to-face by interviewers [26]. Food items were recorded if eaten at least twice a month. For each item, the weekly frequency of consumption, preparation mode and usual portion size were noted. Variations between working days and weekends, as well as seasonal differences in the model of consumption, were considered [26].

### 2.3. Sociodemographic and Lifestyle Information 

In order to collect information on sociodemographic and lifestyle factors, educational level, tobacco use, physical activity, reproductive history, and medical backgrounds, additional questionnaires were administered [26,27]. Trained nurses conducted anthropometric measurements, such as weight and height, during the recruitment process.

### 2.4. Estimation of Nitrosyl-Heme and Heme Iron Content in Processed Meats and Assessment of the Dietary Intakes in EPIC-Spain Cohort

In this study, the term ’meat derivative’ (MD) includes the definition of meat preparations (meat products that still maintain the structure of the meat, even when ground, such as “*hamburguesa*” or “*chorizo fresco*”) and the definition of meat products (referring to meat which has undergone higher processing –in the form of cooking, fermentation or other), such as “*Jamón cocido/jamón de York*” or “*salami*”. This definition is provided by EU legislation [28]. Hereafter, we will use this term to refer to these types of products. The food consumption data of the EPIC-Spain cohort was used to compile a list of MDs.

Due to the highly inaccurate translations and the impracticality of using sausage descriptions, we decided to keep their original names in Spanish. The list of analyzed MD items (n = 52), along with their respective local names, English descriptions, and types of meat used in their production, are shown in Table 1.

Nitrosyl-heme and heme iron contents present in MDs were determined by the High-Performance Liquid Chromatography (HPLC) method in 39 items and analyzed by the Institute of Agrifood Research and Technology (IRTA). Convenience samples were obtained from delicatessen stores, food markets and supermarkets in different geographical locations of Spain and immediately transported to IRTA’s laboratory under refrigerated conditions when required. At least four brands were collected for each product type, ground, mixed, and vacuum-packaged at −20 °C. We conducted analyses on various types of “*chorizo*” and “*morcilla*” samples. Additionally, we examined “*jamón serrano*” and “*jamón cocido/jamón de York*” purchased in different conditions: sliced at butcher shops and pre-packaged in an industrial vacuum or modified atmosphere packaging (n = 4 for each condition). These samples were analyzed separately to evaluate differences in variability between samples and retail packaging conditions. For “*chistorra*” sausages, we divided them into two categories based on the presence or absence of nitrifying agents and analyzed them separately. It is important to highlight that during the 1990s, “*chistorra*” was commonly prepared using nitrifying agents, but their usage has declined in recent times. As a result, factors such as alterations in ingredients or processing methods were taken into account to ensure the representativeness of current samples. Additionally, we included two additional categories: “*lacón tradicional*” (a regional product) and “*lacón industrial*” (more widespread nowadays). These categories were also analyzed separately. Moreover, certain products were analyzed before and after cooking using common Spanish household methods (pan-frying with minimal olive oil).

For the determination of NO-heme and total heme pigment concentrations, we followed the procedure described in Bou et al. (2024) [29]. These pigments were extracted in triplicates using final solutions of 80% acetone and 80% acidified acetone, respectively, considering the sample’s moisture as described by Hornsey (1956) [30]. Briefly, for NO-heme and heme determination, 2 g of ground samples were weighed under subdued light conditions into 50 mL capacity centrifuge tubes. Aliquots of the aqueous and acidified acetone extracts were injected (40 μL) into an Agilent 1100 series HPLC system (Waldbronn, Germany) equipped with a Synergi Fusion-RP column (150 × 4.6 mm, 4 μm, 80 Å) from Phenomenex (Torrance, CA, USA) and using a UV/Vis detector set at 400 and 414 nm. Water with 0.05% trifluoroacetic acid and acetonitrile with 0.05% trifluoroacetic acid were used as mobile phases A and B, respectively. In the aqueous acetone extracts, porphyrins were eluted with a gradient in which phase B increased from 20% to 70% in 5 min and then increased to 100% in 15 min at a constant flow rate of 1 mL min^−1^. In the acidified acetone extract, total heme was eluted with a gradient in which phase B increased from 20% to 70% in 5 min and then increased to 100% in 10 min at a constant flow rate of 1 mL min^−1^.

Nitrosyl-heme and heme iron dietary intake were calculated by multiplying data on nitrosyl-heme and heme iron in MDs by the mean daily intake of related food items for each subject of the cohort. Dietary intakes of nitrosyl-heme and heme iron were based on previously described laboratory determinations available for most consumed EPIC items (39 MDs, 75% of total). For the rest (n = 13), classified as “fresh” MDs (“*albóndigas*”, “*hamburguesas*”, “*salchichas*”, etc.), the dietary intakes of both compounds were indirectly estimated. This approach was chosen as these samples were not considered nutritionally significant sources, with zero or near-zero values, as explained below. A total of 52 items were included in this study to estimate the nitrosyl-heme and heme dietary iron intakes. 

For 25% of MD items, heme iron values were estimated, assuming that 40% of total iron was heme iron. Ingredients with non-heme sources were considered, utilizing data from scientific literature [31,32,33]. For those items, nitrosyl-heme estimates fell into two categories: (1) estimated as zero for MDs lacking nitrifying salts or without high nitrate content (“*albóndigas*” or “*hamburguesas*”), and (2) calculated based on recipes and the nitrosyl-heme content of individual ingredients, such as “*croquetas de carne sin especificar*”.

In the EPIC-Spain study, some records did not specify the types of MDs consumed. These unspecified records were originally grouped together under the label “Not specified (n.s.) MDs”. This aggregated item consumption stood at the beginning of the ranking for most consumed items, both by sex and in all regions of Spain. In order to avoid this, we decided to disaggregate them for this study. For that, we used weighing factors, eliminating the “Not specified (n.s.) MDs” from the list and distributing its consumption among the rest of the MDs. 

Spanish MDs information regarding heme iron and nitrosyl-heme was added to the EPIC-Spain food composition database [34]. 

### 2.5. Statistical Analysis 

Nitrosyl-heme and heme iron were expressed in hemin form (651.94 g/mol molecular weight). Because nitrosyl-heme and heme iron content were right-skewed, we applied the natural logarithm transformation to conduct a symmetric and normal distribution. Arithmetic means for the log-transformed dietary intakes were later exponentiated to obtain the geometric means and the standard deviation. Means were reported as µg/day. Based on descriptive analysis, linear regression analysis (LRA) included the following variables: sex, age, center, BMI (≤25, 25–30, ≥30 kg/m^2^), waist-to-hip ratio (WHR) (low: <0.71 for women and <0.78, for men; normal: ≥0.71 to <0.84 for women and ≥0.78 to <0.94 for men; and high, ≥0.84 for women and ≥0.94 for men), physical activity level (inactive, moderately inactive, moderately active, active) [35], educational level (none, primary school, technical or professional education, secondary school, longer education and not specified) smoking (non-smoker, smoker, ex-smoker and not specified) and energy intake.

The Spearman correlation coefficient between daily consumption of MDs and daily intake of nitrosyl-heme and heme iron was calculated in order to quantify the strength of the linear relationship between the variables. 

In our study of 38,471 subjects, we conducted analyses to identify the major contributors to nitrosyl-heme and heme iron dietary intake. For both components, we calculated the per-item consumption and ranked items by their contribution, from highest to lowest. Additionally, we assessed the influence of different sociodemographic and lifestyle categories on log-transformed nitrosyl-heme and heme iron dietary intakes using *t*-tests, adjusting for sex, age, BMI, center, and energy intake. We also ranked items based on the proportion of variance (R^2^); they explained nitrosyl-heme and heme iron dietary intake to determine the most influential MD by using LRA with stepwise forward selection. Dietary items were included in the regression if they met the 0.05 significance level [36]. In order to carry out this analysis, the logarithms of all items were performed to work under the assumption of normality. Moreover, “*chóped*/*chopped*” was excluded to avoid collinearity. 

All the analyses were conducted using RStudio (4.1.3) with a significance level of *p* < 0.05. 

## 3. Results

Table 2 ranks the most consumed MD items in EPIC Spanish centers, showing their nitrosyl-heme and heme iron contents. Nitrosylation percentages varied by item, reflecting their curing efficiency.

Table 3 shows the top 10 most consumed items by EPIC centers. Differences in MD items and their amounts are noticeable among centers. In parallel, some MD items were consumed in all of the centers. In this regard, 30% of MD items were consumed in the five EPIC-Spain centers; 20% of MD items were consumed in four EPIC-Spain centers, and 30% of the MD items were consumed in three EPIC-Spain centers. MD intake was lowest in Granada (31.7 g/day) and highest in Navarra (45.4 g/day), with significant differences. Overall, our results showed that this Spanish cohort consumed almost 40 g/day of MDs, with a mean daily intake of 7.14 g/day and nearly 79.0 g/day, in the 10 and 90 percentiles, respectively.

With reference to the correlation coefficient between daily consumption of MDs and daily intake of nitrosyl-heme in MD, a very strong correlation was observed (*p* < 0.05). In relation to heme iron from MD, the correlation with MDs was lower but still statistically significant and considered strong (*p* < 0.05).

With regard to the contribution of nitrosyl-heme dietary intake in MDs for the EPIC-Spain cohort (Table 4), five items accounted for nearly 25% of the cohort’s nitrosyl-heme dietary intake from MDs. The items explaining most of the variance in nitrosyl-heme were “*jamón serrano*” (17.6%), “*jamón cocido/jamón de York*” (20.4%), and “*chorizo curado*” (21.9%) when all MDs were considered (expressed as cumulative R2). For heme iron intake, ten items represented just over 30% of the cohort’s heme iron dietary intake, with “*chorizo curado*” contributing the most to the variance (8.7%). 

Table 5 shows the adjusted geometric means of nitrosyl-heme and heme iron separately for categories of sex, age, center, WHR, BMI, smoking status and physical activity. In the Spanish cohort, adjusted geometric means were 528.6 µg/day for nitrosyl-heme and 1676.2 µg/day for heme iron, considering adjustments for sex, age, BMI, center, and energy intake. A significant pattern of differences was found for the daily intake of nitrosyl-heme across the categories of sex, center, energy and educational level. For heme iron, significant differences were found by sex, center, energy, and smoking status. 

The daily geometric mean intake of nitrosyl-heme from MD was highest in men, adults aged 44 to 54 years, overweight individuals, those with high WHR, former smokers and those with longer education. Nitrosyl-heme daily intake consumption seemed to increase with higher levels of physical activity (58.60% difference between active and inactive groups). As to the regional distribution, Navarra had the highest nitrosyl-heme intake, while Granada had the lowest (*p* < 0.05). The same pattern was found for daily heme iron intake, except that, unlike nitrosyl-heme, higher intakes were observed in individuals with technical/professional education and smokers.

The complete list of MDs with nitrosyl-heme and heme iron content is in Appendix A. “*Sangrecilla*” was the item with the highest heme iron content even though its content of nitrosyl-heme was not high. Following in the ranking of items with the highest content of heme iron, we found the different types of “*morcillas*”: “*morcilla de cebolla*”, “*morcilla de arroz*”, “*morcilla asturiana*” and “*morcilla murciana*”. In regard to the content of nitrosyl-heme, “*morcilla asturiana*” was the item with the highest values, which differentiates it from other “*morcillas*”, due to the specific methods of smoking, drying, and fermentation utilized in their production. Next were “*cecina*”, “*paté/fuagrás/foie-gras*”, “*frankfurt*” and “*chorizo tipo vela*”.

We also calculated the average consumption per item, given the fact that some MDs may not be high in nitrosyl-heme or heme iron content but still be highly consumed by the EPIC-Spain cohort. This is what happens, for instance, with “*jamón serrano*”, “*jamón cocido/jamón de York”,* and “*chorizo curado*”. These are all low nitrosyl-heme-containing foods, but they are highly consumed by this Spanish population. Conversely, some items present a high content of nitrosyl-heme, but their daily intakes in the EPIC-Spain cohort were low. Examples of this include “*cecina*”, “*chorizo tipo vela*”, or “*longaniza imperial*” (Appendix A). The same pattern was observed for heme iron content with “*jamón cocido/jamón de York*”, “*croquetas de carne sin especificar, caseras*”, and “*jamón serrano*”, which had low heme iron content but remained at high daily intake by the EPIC-Spain cohort.

Appendix A shows the ranking of the most consumed MD items that contribute the most to total nitrosyl-heme dietary intake for EPIC-Spain cohort. “*Jamón serrano*” had the highest contribution, followed by “*jamón cocido/jamón de York*” and various “*chorizo*” types. When considering only the consumers of those items, we observed significant variations in the results, with ”*carne en lata*” as the top contributor (1038.22 µg/d), followed by “*frankfurt*” (692.0 µg/d), “*cecina*” (585.9 µg/d) and “*chorizo curado*” (524.2 µg/d).

Regarding the items that contribute the most to total heme iron dietary intake to EPIC-Spain cohort, Appendix A shows that “*albóndigas de carne sin especificar caseras*”, “*jamón serrano*”, “*morcilla de cebolla*”, and “*jamón cocido/jamón de York*”, were the highest contributors. Some MD items, such as “*hamburguesa de carne sin especificar*”, “*cecina*”, or “*sangrecilla*”, contained large amounts of heme iron, but their contribution in the cohort was low. When considering only the consumers of those items, it was observed that “*sangrecilla*” was the leading contributor to the total dietary intake of heme iron (16,612.3 µg/d), followed by “*morcilla de arroz*” (13,075.8 µg/d) and “*morcilla de cebolla*” (6208.3 µg/d).

## 4. Discussion

To the best of our knowledge, no data on the levels of nitrosyl-heme consumption for any population were available prior to this study. Moreover, any available information on nitrosyl-heme and heme iron content in MDs was very scarce [15,16,17,18,19]. Our study is the first to present data based on direct content measurements by HPLC in MDs, and it does so for a large (n = 38,471) prospective cohort. 

Among the top 10 globally consumed MD items in the EPIC Spanish population, “*jamón serrano*” *and* “*jamón cocido/jamón de york*” were the most popular. In contrast, different trends were observed in other populations, even when they are similar, such as Italy or Germany where the most consumed MD items were dry-fermented sausages [37] and bratwurst [38], respectively.

Regarding differences between EPIC centers our study shows that the consumption of several MDs was variable, possibly because of local gastronomic differences. Although the 10 most-consumed items were mostly the same for all centers (but with varying rankings of consumption), some centers such as Murcia, Asturias or Gipuzkoa presented typical regional foods (like “*morcilla murciana*”, “*lacón*” and “*morcilla de cebolla*”, respectively) amidst one of their most consumed MD items. Even though the percentage of non-consumers for some MDs is very high, they still stand within the top 10 of most consumed MD items. This evidence suggests high dietary intakes for those individuals who indeed consumed them.

With regard to sociodemographic and lifestyle characteristics, once adjusted for energy intake, males reported higher daily intakes of nitrosyl-heme and heme iron compared to females. This could be attributed to men generally consuming more meat and MDs [39,40], both rich in nitrosyl-heme and heme iron. Nowadays, the tendency for higher dietary intake of MDs by men appears to be extensive globally, as has already been reported in Europe, Australia and the USA [41]. In our study, “*jamón cocido/jamón de York*” was the most-consumed MD item among women, while “*jamón serrano*” was preferred by men. These results seem to be different compared to those in other populations [41,42].

As anticipated, a notably positive correlation was observed between MDs and nitrosyl-heme dietary intake. This strong positive correlation can be attributed to the analysis of foods known to be rich in heme iron/nitrosyl-heme, exclusively considering heme iron/nitrosyl-heme derived from these food sources. Furthermore, these findings align with expectations, given that MDs represent the sole source of nitrosyl-heme within the diet [43], while heme iron is derived from a broader range of protein foods [44]. Considering these correlations, we compared our results with those of Rouhani et al. 2014 [45], which showed a similar association between red meat and MD consumption and obesity risk, higher BMI, and waist circumference. While our findings align in the same direction, they were not statistically significant (*p* > 0.05). 

The items that contributed the most to dietary intakes of nitrosyl-heme and heme iron were also identified. It explains how foods contribute to their total dietary intake and to between-person variance. In this study, five food items alone explained 24.5% of nitrosyl-heme intake variance. For heme iron, nine MD items explained almost 30% of the variance. Contrary to findings in other studies, where fewer items are needed to explain a given proportion of between-person variance in a single nutrient/component than the same proportion of between-person variance in total intake [46,47], our study presents a distinct pattern. As observed, more than five items are needed for this purpose.

Regarding the amount of nitrosyl-heme and heme iron present in each of the MDs, we categorized and ordered those items from highest to lowest mean daily intake (g/day), presenting their amounts per gram of MDs. There is only one literature source from which we are able to compare our results on curing efficiency. It is a single study that estimated the nitrosylated heme iron content for each type of MDs by applying a fixed value provided by the French Pork Institute [20], resulting in higher curing efficiency (67%) than our study, with which average curing efficiencies were 44% for raw MDs and 52% for cooked MDs using HPLC data.

Referring to the curing efficiency, some items presented 100% or more of nitrosylated heme iron due to analytical error with random components, producing random variations in the data, but when analyzed, they tend to compensate for each other. The mean error tended to be 0, but the quantities were not to induce bias that could create a random error.

Our results showed an appreciable variability in the proportion of nitrosylated heme iron, dependent on the type of MDs. For example, referring to “*sangrecilla*” was the item with the highest heme iron content because it is made from coagulated blood. Still, its nitrosylation percentage is one of the lowest, so its contribution to the dietary intake of nitrosyl-heme was unexpected since there is no added nitrifying agents in their formulation. “*Morcilla asturiana*” stood out with higher values of nitrosyl-heme due to unique smoking, drying, and fermentation methods in its production, setting it apart from other “*morcillas*” and showing a 55% nitrosylation. Conversely, ”*cecina*” had high nitrosylation, likely due to its high heme iron (183 mg/kg) and nitrosyl-heme (105 mg/kg) content, resulting from rich protein and heme proteins in beef meat.

The proportion of nitrosylated heme iron could potentially elucidate the conversion of nitrate and/or nitrite into NO, resulting in an elevated nitrosyl-heme content while also providing insights into the equilibrium and stability of various heme species under different conditions. In this study, we present the nitrosyl-heme and heme iron determination using the results obtained from the HPLC method. We did so to avoid potential overestimations of some MDs if the Hornsey (classical spectrophotometric) method had been used. This finding could provide an explanation for curing efficiencies that surpass 100% in products that include paprika as an ingredient due to induced analytical error, which resulted in random variations in the data.

We calculated the average consumption per item due to varying nitrosyl-heme content. Some low-nitrosyl-heme foods like “*jamón serrano*” and “*jamón cocido/jamón de York*”, were highly consumed in this cohort. Conversely, some items with high nitrosyl-heme had low intakes (e.g., “*cecina*” or “*chorizo tipo vela*”*).* Notably, for consumers of those items, “*carne en lata*”, “*cecina*”, *and* “*sangrecilla*”, usually low contributors in the cohort, ranked high, indicating significant consumption. 

A similar situation was observed for heme iron content; “*jamón cocido/jamón de York*”, “*croquetas de carne sin especificar*, *caseras*” or “*jamón serrano*” presented low heme iron contents but high daily intake by the EPIC-Spain cohort. In contrast, MDs like “*hamburguesa de carne sin especificar*”, “*cecina*” *or* “*sangrecilla*” contained significant heme iron amounts, but their contribution in the cohort was low. Concerning the top contributors to heme iron intake among consumers of those items, there was variability. “*Sangrecilla*” topped the list, followed by “*morcilla de arroz*” and “*morcilla de cebolla*”. Conversely, “*jamón cocido/jamón de York*” or “*chorizo curado*” made smaller contributions to heme iron intake. This highlights that consumers of those items, which are less commonly consumed overall, consumed significant quantities of them.

It is important to note that most studies presented PM dietary intake data as a global value and did not differentiate items by their subtype, cooking or curing processes, or added ingredients [48,49,50]. Quantifying the mean consumption of different PMs, as well as their treatments, cooking and added ingredients, is important since differences in processing and composition could be associated with health outcomes [51]. To date, very few studies have assessed PMs according to their subtypes [41,52,53]. This is the reason why one of our research goals was to identify the different types of PMs consumed in the EPIC-Spain cohort and to analyze their composition. Furthermore, up to the present time, not many studies have assessed the combined analysis of nitrosyl-heme and heme iron content in MDs [54,55], making our research valuable in this context. 

An important limitation to our analysis is that the Spanish EPIC study was based on a non-representative sampling of the Spanish general population. However, the prospective design precluded recall bias, while selection bias was minimized by the very high rate of follow-up over a long period of time. Moreover, subjects came from different socio-economic backgrounds and different geographical regions, and the pattern of dietary intake and other lifestyle factors was very similar to those observed in population surveys performed in the Spanish areas included in the present study [51,56]. Regarding the strengths of the study, dietary intakes of nitrosyl-heme and heme iron contents were calculated based on laboratory measurements of selected food items using the HPLC method. We also observed a very large dataset (38,471 subjects); a large number (n = 52) of MD items were included, most of them analyzed (n = 39) and used through validated methodology in the dietary estimation [57]. Furthermore, the MD samples analyzed were obtained from different geographical locations, including different elaboration methods, commercial brands and packaging, and the changes in manufacturing that have occurred from the 90s until now have been taken into account. This is important considering that even high-quality food composition tables will produce poor estimates when combined with low-quality data because of the lack of validity and/or reliability of a dietary questionnaire.

## 5. Conclusions

In conclusion, we have assessed nitrosyl-heme iron and heme iron intake in the EPIC-Spain cohort, identifying the major contributors and exploring sociodemographic and lifestyle factors. This contributes valuable insights into nitrosyl-heme exposure, particularly its association with processed meat and colorectal cancer risk. 

## Figures and Tables

**Table 1 nutrients-16-00878-t001:** List of meat derivatives analyzed, with their local product name and their English description and type of meat.

Local Product Name	English Description + Meat
Albóndigas de carne sin especificar	Meatballs, meat not specified
Albóndigas de carne sin especificar, caseras	Meatballs, meat not specified, homemade
Albóndigas de cerdo	Meatballs, pork
Albóndigas de ternera	Meatballs, veal
Bacon	Cured pork bacon
Bratwurst	Bratwurst type pork sausage
Cabeza de jabalí	Pork head cheese
Carne en lata	Canned pork meat/corned beef
Cecina	Dry-cured beef leg
Chicharrones	Greaves, pork
Chistorra	Small caliber raw pork paprika sausage
Chóped/Chopped	Pork meat luncheon/jagdswurst
Chorizo curado	Fermented dry-cured pork paprika sausage
Chorizo fresco oreado	Aired raw pork paprika sausage
Chorizo de Pamplona	Fermented dry-cured pork paprika sausage, Pamplona type
Chorizo tipo vela	Fermented dry-cured pork paprika sausage, Revilla type
Croquetas de carne sin especificar	Croquettes, meat not specified
Fiambre de pavo	Turkey meat luncheon
Frankfurt	Frankfurter type sausage (chicken and/or pork)
Fuet	Small caliber dry-cured pork sausage, Catalan specialty
Hamburguesa de carne sin especificar	Hamburguer, meat not specified
Hamburguesa de carne sin especificar, bar/restaurante	Hamburguer, meat not specified, bar/restaurant
Hamburguesa de carne sin especificar, casera	Hamburguer, meat not specified, homemade
Hamburguesa de pollo	Hamburguer, chicken
Hamburguesa de ternera	Hamburguer, veal
Jamón cocido/Jamón de York	Cooked pork ham
Jamón serrano	Dry-cured pork ham
Lacón, industrial	Cooked pork ham, industrial elaboration, Galician specialty
Lacón, tradicional	Cooked pork ham, traditional elaboration, Galician specialty
Lomo embuchado	Dry-cured pork loin
Longaniza de payés/llonganissa de pagés	Dry-cured pork sausage, Catalan specialty
Longaniza imperial	Fermented dry-cured pork sausage, Murcian specialty
Longaniza murciana/salchicha de pellizco	Dry-cured pork sausage, Murcian specialty
Longaniza seca	Dry-cured pork sausage
Morcilla asturiana	Smoked pork blood sausage, Asturian specialty
Morcilla de arroz	Pork blood sausage with rice
Morcilla de cebolla	Pork blood sausage with onion
Morcilla murciana	Pork blood sausage Murcian specialty
Morcón	Bung cap fermented dry-cured pork paprika sausage
Morcón blanco	Pork bung cap sausage
Morcón con sangre	Pork bung cap blood and liver sausage
Mortadela	Pork mortadella
Panceta de cerdo	Dry-cured pork belly
Paté/Fuagrás/Foie-gras	Pork liver paté/duck liver foie gras
Relleno	Pork sausage, with egg, rice and onion
Salami	Fermented pork salami
Salchicha fresca del país	Sausage, country style, fresh
Salchicha de pollo	Chicken sausage, fresh
Salchicha fresca murciana	Pork sausage, Murcian specialty, fresh
Salchichón	Dry-cured pork sausage
Sangrecilla	Cooked blood curd (chicken, pork or veal)
Sobrasada/sobrassada	Cured pork paprika sausage and spread
Tocino	Salted pork backfat

**Table 2 nutrients-16-00878-t002:** Top 10 most consumed meat derivative items globally and nitrosyl-heme and heme iron content measured by HPLC method.

Items	Mean (g/d)	NOheme * (µg/g) **	Heme Iron (µg/g) **	Curing Efficiency (%)
Jamón serrano	5.30	39	41	95.1
Jamón cocido/jamón de York	5.04	22	27	81.5
Albóndigas de carne sin especificar, caseras	2.62	0 ***	105	0.0
Croquetas de carne sin especificar	1.70	1 ****	1	100.0
Chorizo curado	1.67	53	53	100.0
Tocino	1.33	1	1	100.0
Panceta de cerdo	1.11	6	17	35.3
Chorizo fresco oreado	1.01	31	29	106.9
Albóndigas de ternera	0.88	0 ***	143	0
Salchichón	0.58	44	56	78.6

* NOheme—Nitrosyl-heme. ** data expressed as hemin (651.94 g/mol). *** Values estimated as zero. **** Value computed using the nitrosyl-heme content of ingredients.

**Table 3 nutrients-16-00878-t003:** Top 10 most consumed items by EPIC-Spain center (Asturias, Navarra, Granada, Murcia, and Gipuzkoa) (Mean intake + SD).

EPIC-Center	% Non Consumers	Items	Mean (g/d)	SD
Asturias	55.6	Jamón cocido/jamón de York	3.21	7.4
	58.9	Jamón serrano	2.93	7.0
	62.7	Chorizo curado	2.76	6.0
	23.2	Tocino	2.54	2.9
	41.4	Lacón	2.15	2.9
	88.2	Albóndigas de carne sin especificar, caseras	1.85	5.9
	84.7	Croquetas de carne sin especificar	1.75	5.0
	96.1	Chorizo fresco oreado	1.16	7.0
	79.3	Panceta de cerdo	0.70	1.8
	58.3	Morcilla asturiana	0.57	0.9
Granada	22.1	Jamón serrano	7.28	7.0
	40.2	Jamón cocido/jamón de York	6.80	7.4
	61.7	Salchichón	2.00	1.5
	84.9	Croquetas de carne sin especificar	1.76	5.0
	91.3	Albóndigas de carne sin especificar, caseras	1.25	5.9
	54.6	Tocino	1.26	2.9
	94.2	Frankfurt	0.42	2.4
	94.8	Chorizo fresco oreado	0.37	7.0
	96.4	Chóped/Chopped	0.36	1.1
	97.6	Albóndigas de cerdo	0.34	0.7
Murcia	24.5	Jamón serrano	6.46	7.0
	26.7	Jamón cocido/jamón de York	5.57	7.4
	68.6	Croquetas de carne sin especificar	1.12	5.0
	59.8	Morcilla murciana	1.06	0.0
	95.8	Albóndigas de carne sin especificar	1.00	1.4
	82.0	Salchicha fresca murciana	0.56	0.0
	32.0	Tocino	0.45	2.9
	39.0	Longaniza de payés	0.42	0.0
	89.7	Frankfurt	0.30	2.4
	91.3	Longaniza imperial	0.26	0.1
Navarra	28.8	Jamón serrano	6.88	6.4
	45.0	Jamón cocido/Jamón de York	5.36	7.4
	70.1	Albóndigas de carne sin especificar, caseras	4.46	5.9
	34.0	Panceta de cerdo	3.88	1.8
	84.3	Albóndigas de ternera	2.42	2.7
	75.1	Chistorra	2.04	0.0
	79.3	Croquetas de carne sin especificar	2.91	5.0
	82.7	Chorizo curado	1.86	6.0
	57.1	Tocino	1.74	2.9
	89.7	Salchicha del país	1.02	1.2
	91.8	Chorizo de Pamplona	0.73	2.6
Gipuzkoa	62.6	Albóndigas de carne sin especificar, caseras	5.44	5.9
	56.0	Jamón cocido/Jamón de York	4.42	7.4
	64.4	Chorizo curado	3.60	6.0
	61.8	Chorizo fresco oreado	3.39	7.0
	57.2	Jamón serrano	3.11	7.0
	79.6	Croquetas de carne sin especificar	1.96	5.0
	89.4	Hamburguesa de carne sin especificar, casera	1.93	2.9
	90.1	Albóndigas de ternera	1.31	2.7
	88.1	Morcilla de cebolla	1.14	0.0
	85.7	Panceta de cerdo	0.78	1.8

**Table 4 nutrients-16-00878-t004:** Contribution to the variability of nitrosyl-heme and heme iron based on meat derivatives in the EPIC-Spain cohort.

	Contribution to Total Intake of Nitrosyl-Heme		Contribution to Total Intake of Heme Iron
Food Item	Adj. R2	Food Item	Adj. R2
LRA 1: adjusted by Jamón serrano	0.176	LRA 1: adjusted by Chorizo curado	0.087
LRA 2: LRA 1 + Chorizo curado	0.204	LRA 2: LRA 1 + Albóndigas de carne sin especificar, caseras	0.124
LRA 3: LRA 2 + Jamón cocido/Jamón de York	0.219	LRA 3: LRA 2 + Jamón serrano	0.160
LRA 4: LRA 3 + Tocino	0.234	LRA 4: LRA 3 + Lacón	0.195
LRA 5: LRA 4 + Albóndigas de carne, sin especificar	0.245	LRA 5: LRA 4 + Albóndigas de ternera	0.225
LRA 6: LRA 5 + Chorizo fresco oreado	0.252	LRA 6: LRA 5 + Morcilla de cebolla	0.249
LRA 7: LRA 6 + Lacón	0.260	LRA 7: LRA 6 + Longaniza de payés/llonganissa de pagés	0.264
LRA 8: LRA 7 + Chorizo tipo Pamplona	0.267	LRA 8: LRA 7 + Salchichón	0.279
LRA 9: LRA 8 + Sobrasada/sobrassada	0.275	LRA 9: LRA 8 + Hamburguesa de carne sin especificar, casera	0.294
LRA 10: LRA 9 + Frankfurt	0.282	LRA 10: LRA 9 + Albóndigas de carne sin especificar	0.306
LRA 11: LRA 10 + Longaniza de payés/llonganissa de pagés	0.286	LRA 11: LRA 10 + Chorizo fresco oreado	0.318
LRA 12: LRA 11 + Panceta de cerdo	0.291	LRA 12: LRA 11 + Morcilla asturiana	0.329
LRA 13: LRA 12 + Lomo embuchado	0.295	LRA 13: LRA 12 + Morcilla murciana	0.339
LRA 14: LRA 13 + Salchichón	0.298	LRA 14: LRA 13 + Jamón cocido/Jamón de York	0.348
LRA 15: LRA 14 + Chistorra	0.306	LRA 15: LRA 14 + Morcilla de arroz	0.357
LRA 16: LRA 15 + Morcilla asturiana	0.305	LRA 16: LRA 15 + Hamburguesa de ternera	0.366
LRA 17: LRA 16 + Paté/Fuagrás/Foie-gras	0.308	LRA 17: LRA 16 + Frankfurt	0.369
LRA 18: LRA 17 + Albóndigas de ternera	0.310	LRA 18: LRA 17 + Sobrasada/sobrassada	0.374
LRA 19: LRA 18 + Hamburguesa de carne sin especificar, casera	0.313	LRA 19: LRA 18 + Salchicha fresca del país	0.379
LRA 20: LRA 19 + Morcilla de cebolla	0.314	LRA 20: LRA 19 + Sangrecilla	0.383
LRA 21: LRA 20 + Morcilla murciana	0.315	LRA 21: LRA 20 + Lomo embuchado	0.387
LRA 22: LRA 21 + Croquetas de carne sin especificar	0.316	LRA 22: LRA 21 + Panceta de cerdo	0.389
LRA 23: LRA 22 + Chorizo tipo vela	0.317	LRA 23: LRA 22 + Albóndigas de cerdo	0.392
LRA 24: LRA 23 + Sangrecilla	0.318	LRA 24: LRA 23 + Chorizo de Pamplona	0.394
LRA 25: LRA 24 + Longaniza murciana/salchicha de pellizco	0.319	LRA 25: LRA 24 + Hamburguesa de pollo	0.396
LRA 26: LRA 25 + Longaniza imperial	0.319	LRA 26: LRA 25 + Paté/Fuagrás/Foie-gras	0.398
LRA 27: LRA 26 + Morcilla de arroz	0.319	LRA 27: LRA 26 + Hamburguesa de carne sin especificar, bar/restaurante	0.400
LRA 28: LRA 27 + Bacon	0.320	LRA 28: LRA 27 + Chóped/Chopped	0.402
LRA 29: LRA 28 + Hamburguesa de ternera	0.320	LRA 29: LRA 28 + Tocino	0.403
LRA 30: LRA 29 + Salami	0.320	LRA 30: LRA 29 + Relleno	0.403
LRA 31: LRA 30 + Longaniza seca	0.320	LRA 31: LRA 30 + Longaniza murciana/salchicha de pellizco	0.404
LRA 32: LRA 31 + Cecina	0.320	LRA 32: LRA 31 + Salchicha fresca murciana	0.405
LRA 33: LRA 32 + Embutido de pavo	0.320	LRA 33: LRA 32 + Chorizo tipo Vela	0.405
LRA 34: LRA 33 + Carne en lata	0.321	LRA 34: LRA 33 + Salami	0.406
LRA 35: LRA 34 +Salchicha de pollo	0.321	LRA 35: LRA 34 + Longaniza imperial	0.406
LRA 36: LRA 35 + Albóndigas de carne sin especificar	0.321	LRA 36: LRA 35 + Cecina	0.406
LRA 37: LRA 36 + Mortadela	0.321	LRA 37: LRA 36 + Salchicha de pollo	0.407
LRA 38: LRA 37 + Hamburguesa de carne sin especificar, bar/restaurante	0.321	LRA 38: LRA 37 + Croquetas de carne sin especificar	0.407

*p*-value for entrance forward stepwise regression < 0.05. LRA—Linear regression analysis.

**Table 5 nutrients-16-00878-t005:** Adjusted nitrosyl-heme and heme iron intakes by lifestyle and sociodemographic characteristics in EPIC-Spain participants.

			Nitrosyl-Heme (µg/d) §**			Heme Iron (µg/d) ✣**		
		n	Geometric Mean	SD	Median	Geometric Mean	SD	Median
**Total ***		38,471	529	470.9	397	1676	1057.9	1396
**Sex** §✣								
Male		14,632	705	632.2	534	2267	1312.6	1936
Female		23,839	420	305.5	344	1312	650.6	1165
**Age (y)**								
<33		16	533	236.0	456	1504	694.6	1238
33 to <44		11,689	528	524.3	406	1624	979.2	1361
44 to <54		15,420	550	558.5	411	1764	1110.2	1462
54 to < 64		10,371	492	494.2	375	1604	999.3	1342
≥64		975	421	343.1	335	1347	786.1	1126
**BMI (kg/m^2^)**								
<25		8533	493	394.9	398	1644	890.1	1429
>25–30		18,549	546	474.3	411	1749	1079.0	1460
>30		11,389	527	516.1	377	1579	1115.8	1261
**Center** §✣								
Asturias		8052	418	250.9	351	1466	785.3	1256
Granada		7223	356	238.9	294	1255	746.0	1061
Murcia		7956	509	354.6	412	1740	1065.4	1449
Navarra		7638	567	389.5	452	1942	1185.0	1604
Gipuzkoa		7602	527	342.2	430	1839	1066.1	1554
**WHR (cm)**								
Low		531	402	285.8	336	1410	745.6	1247
Normal		19,650	512	434.4	395	1702	1081.8	1407
High		18,290	550	510.0	403	1781	1265.7	1437
**Cambridge Physical Activity Index (Met-h/week)**								
Inactive		14,771	459	380.0	361	1397	787.8	1183
Moderately inactive		12,773	508	447.1	391	1584	930.7	1325
Moderately active		6799	596	533.8	446	1968	1186.2	1663
Active		4128	728	629.6	555	2246	1292.6	1937
**Level School** §								
None		13,110	486	435.9	362	1455	919.2	1178
Primary School		14,891	526	483.4	392	1712	1055.8	1401
Technical, Professional		3180	573	502.9	431	1985	1146.8	1690
Secondary School		2525	591	504.3	467	1776	1063.5	1504
Longer Education		4509	598	476.2	479	1725	931.1	1478
Not specified		256	491	341.6	404	1481	772.5	1280
**Smoking Status** ✣								
Never		21,211	448	372.8	348	1420	835.1	1180
Former		6815	641	557.5	494	1918	1094.8	1625
Current		10,423	620	459.4	471	1948	1169.1	1633
Not specified		22	518	581.3	242	1476	841.3	1330

* Adjusted by sex, age, BMI, center and energy intake. § *p*-value was significant (*p*-value < 0.05) in sex, energy, center and level school for nitrosyl-heme intake ✣ *p*-value were significant (*p*-value < 0.05) in sex, energy, center and smoking status for heme iron intake ** data expressed as hemin (molecular weight: 651.94 g/mol).

## Data Availability

The raw data supporting the conclusions of this article will be made available by the authors on reasonable request. The data are not publicly available due to the research group’s policy.

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
