# Peer review of "Nitrosyl-Heme and Heme Iron Intake from Processed Meats in Subjects from the EPIC-Spain Cohort"

_nutrients, 2024, doi:10.3390/nu16060878_

Round 1
Reviewer 1 Report
Comments and Suggestions for Authors
The manuscript presents interesting data that is of significant interest to society.
However, please check the followings:
- there are phrases that are similar in the abstract and in the introduction of the manuscript (e.q. In 2015, the International Agency for Research on Cancer (IARC) [4] classified PMs as “carcinogenic” ..) please reformulate,
- you mention in the abstract the analysis of the nitrosyl-heme and heme iron levels in the processed meats by HPLC, however there is no method description, did you get the approval to use that data?
- even if you used a method previously described by someone else it is necessary to include the sample size, the sample processing steps to be able to analyze, what type of detector and what kind of LC system was used. Moreover, in the discussion you first mention the quantification by HPLC and then you regard it as one of the strengths of this paper, which make it mandatory for a more complete methodology description.
- phrases such as "More detailed information can be found elsewhere" are not justified unless all authors have been part of the mentioned paper. Since this is not the case, please describe the methods accordingly.
- there is inconsistent formatting through the manuscript (see table 2 title versus table 3/table 4..)
Comments on the Quality of English LanguageThe term "model" is used for the description of various elements and this makes the reading difficult. Please find an appropriate term for statistics that is different from the dietary model, or the other way around
Author Response
Dear reviewer,
Thank you very much for all your comments and suggestions, as well as for taking the time to review this manuscript. Your input has been considered in the final version.
Comments and Suggestions for Authors
Comment 1. There are phrases that are similar in the abstract and in the introduction of the manuscript (e.q. In 2015, the International Agency for Research on Cancer (IARC) [4] classified PMs as “carcinogenic” ..) please reformulate.
Response 1.
Page 1 (line 39-41) We have modified the phrasing of the abstract introduction to avoid duplication with the Introduction section of the manuscript.
"The consumption of processed and red meats is linked to the likelihood of developing colorectal cancer. Various theories have been proposed to explain this connection, with particular focus on nitrosyl-heme and heme iron intake."
Comment 2. You mention in the abstract the analysis of the nitrosyl-heme and heme iron levels in the processed meats by HPLC, however there is no method description, did you get the approval to use that data?
Response 2. Yes, we decided not to mention the HPLC analytical method in the abstract because it was described in a previous paper by our group (Bou et al 2024). However, as you recommend, we added more information about the analytical method in the section "Estimation of nitrosyl-heme and heme iron content in processed meats and assessment of the dietary intakes in EPIC-Spain cohort".
To respond to your latest inquiry, we have obtained approval to utilise this data.
Comment 3. Even if you used a method previously described by someone else it is necessary to include the sample size, the sample processing steps to be able to analyze, what type of detector and what kind of LC system was used. Moreover, in the discussion you first mention the quantification by HPLC and then you regard it as one of the strengths of this paper, which make it mandatory for a more complete methodology description.
Response 3.
Page 5 (line 165-180). As you recommend, we added the sample size, the sample processing steps to be able to analyse, what type of detector and what kind of HPLC system we used. The paragraph would be as follows:
”For the determination of NO-heme and total heme pigment concentrations we followed the procedure described in Bou et al (2024) [30]. These pigments were extracted in triplicates by using final solutions of 80% acetone and 80% acidified acetone, respectively, considering the sample’s moisture as described by Hornsey (1956)[29] Briefly, for NO-heme and heme determination, 2 g of ground samples were weighed under subdued light conditions into 50 mL capacity centrifuge tubes. Aliquots of the aqueous and acidified acetone extracts were injected (40 μL) into an Agilent 1100 series HPLC system (Waldbronn, Germany) equipped with a Synergi Fusion-RP column (150 × 4.6 mm, 4 μm, 80 Å) from Phenomenex (Torrance, USA) and using a UV/Vis detector set at 400 and 414 nm. Water with 0.05% trifluoroacetic acid and acetonitrile with 0.05% trifluoroacetic acid were used as mobile phases A and B, respectively.In the aqueous acetone extracts, porphyrins were eluted with a gradient in which phase B increased from 20% to 70% in 5 min and then increased to 100% in 15 min at a constant flow rate of 1 mL min -1 . In the acidified acetone extract, total heme was eluted with a gradient in which phase B increased from 20% to 70% in 5 min and then increased to 100% in 10 min at a constant flow rate of 1 mL min -1.”
Comment 4. Phrases such as "More detailed information can be found elsewhere" are not justified unless all authors have been part of the mentioned paper. Since this is not the case, please describe the methods accordingly.
Response 4.
Page 3 (line 122). For Diet information section, where we added the phrase "more detailed information can be found elsewhere", and considering that the most crucial information regarding diet collection is covered, we opted to omit that sentence.
Comment 5. There is inconsistent formatting through the manuscript (see table 2 title versus table 3/table 4..)
Response 5. We modified the table format to align with their format for consistency.
Comments on the Quality of English Language
Comment 1. The term "model" is used for the description of various elements and this makes the reading difficult. Please find an appropriate term for statistics that is different from the dietary model, or the other way around
Response 1.
Page 6 and 9. Thank you for the opportunity to improve our manuscript. We decided to use for the dietary model the word “model” and for all statistics associations the word “Linear regression analysis (LRA)”. This last term was included in Table 4.
Reviewer 2 Report
Comments and Suggestions for Authors
This is an intresting study but there are certain issues that the authors should take into consideration.
1. The EPIC study was conducted several years ago and since then there is a growing trend towards "green nitrates". Did they check whether the product they collected for analysis contained sodium nitrate or green nitrates.
2. There were 52 items recorded in Table 1 but only 39 were analysed. My understanding is that the home produced products could not have been analysed since they do not contain added nitrites. However, it should be reported which products were excluded and why.
3. The exact number of products analysed for each category should be reported. Which criteria did you apply in order to collect the samples? Also, there are 5 regions in the study and you only collected 4 samples. Please be more specific in providing information about sampling.
4. Please replace country with region (Table 3)
Author Response
Dear reviewer,
Thank you very much for all your comments and suggestions, as well as for taking the time to review this manuscript. Your input has been considered in the final version.
Comments and Suggestions for Authors
Comment 1. The EPIC study was conducted several years ago and since then there is a growing trend towards "green nitrates". Did they check whether the product they collected for analysis contained sodium nitrate or green nitrates.
Response 1. The label of the studied products indicated the addition of E-249 to 252. Therefore, it is reasonable to think that the producers did not employ “green nitrates” such as celery powder to maintain the red colour of products. However, in some products, the presence of natural sources of nitrates from plants can be considered as normal. These nitrates should be first reduced to nitrites by microorganisms with nitrate-reductase activity as in the case of many Staphylococcus sp.
Comment 2. There were 52 items recorded in Table 1 but only 39 were analysed. My understanding is that the home produced products could not have been analysed since they do not contain added nitrites. However, it should be reported which products were excluded and why.
Response 2.
Page 5-6 (line 185-188). As you mentioned, a sentence has been added reporting which products were not analysed and why.
“For the rest (n=13), classified as "fresh" MDs (“albóndigas”, “hamburguesas” “salcihchas”, etc.), the dietary intakes of both compounds were indirectly estimated. This approach was chosen as these samples were not considered nutritionally significant sources, with zero or near-zero values, as explained below.
Comment 3. The exact number of products analysed for each category should be reported. Which criteria did you apply in order to collect the samples? Also, there are 5 regions in the study and you only collected 4 samples. Please be more specific in providing information about sampling.
Response 3. It is important to mention that most of the studied products are widely consumed in Spain and the same big brands are marketed all over the country. It is also worth noting that many supermarkets and markets sell the same popular brands. Therefore, we considered the supermarkets with the largest share in each region to select 4 brands. We found that the same big supermarkets brands and distributors are present in the different regions and the only difference is their relative position in the rank. Therefore, we considered the top 4 supermarkets for the sampling and when it was not possible to acquire the 4 different samples we decided to go to an alternative supermarket with a relevant share in the region. In the case of certain regional specialities that are not widely marketed, it was difficult to find 4 different brands, and, in this case, we look for specific brands and small producers in local markets and stores.
We added the followed sentences in order to provide more information about the sampling:
Page 5 (line 146-164). “Convenience samples were obtained from delicatessen stores, food markets and supermarkets in different geographical locations of Spain and immediately transported to IRTA’s laboratory under refrigerated conditions, when required. At least 4 brands were collected for each product type, ground, mixed, and vacuum-packaged at -20°C. We conducted analyses on various types of "chorizo" and "morcilla" samples. Additionally, we examined "jamón serrano" and "jamón cocido/jamón de York" purchased in different conditions: sliced at butcher shops, and pre-packaged in industrial vacuum or modified atmosphere packaging (n = 4 for each condition). These samples were analyzed separately to evaluate differences in variability between samples and retail packaging conditions. For "chistorra" sausages, we divided them into two categories based on the presence or absence of nitrifying agents and analyzed them separately. It's important to highlight that during the 1990s, "chistorra" was commonly prepared using nitrifying agents, but their usage has declined in recent times. As a result, factors such as alterations in ingredients or processing methods were taken into account to ensure the representativeness of current samples. Additionally, we included two additional categories: "lacón tradicional" (a regional product) and "lacón industrial" (more widespread nowadays). These categories were also analyzed separately. Moreover, certain products were analysed before and after cooking using common Spanish household methods (pan-frying with minimal olive oil).”
Comment 4. Please replace country with region (Table 3)
Response 4. Thank you very much for the comment and the opportunity to improve our manuscript. It was a mistake, but we have chosen to replace the word "Country" with "EPIC-Centre" to maintain consistency with the text.
Round 2
Reviewer 2 Report
Comments and Suggestions for Authors
-